# Exploring the Role of Hypoxia-Inducible Carbonic Anhydrase IX (CAIX) in Circulating Tumor Cells (CTCs) of Breast Cancer

**DOI:** 10.3390/biomedicines11030934

**Published:** 2023-03-17

**Authors:** Julianne D. Twomey, Baolin Zhang

**Affiliations:** Office of Biotechnology Products, Center for Drug Evaluation and Research, Food and Drug Administration, Silver Spring, MD 20993, USA

**Keywords:** circulating tumor cells (CTCs), cancer stem cells (CSC), breast cancer, biomarker, liquid biopsy, molecular profile, metastasis, cancer therapy

## Abstract

Circulating tumor cells (CTCs) in the peripheral blood are believed to be the source of metastasis and can be used as a liquid biopsy to monitor cancer progression and therapeutic response. However, it has been challenging to accurately detect CTCs because of their low frequency and the heterogeneity of the population. In this study, we have developed an in vitro model of CTCs by using non-adherent suspension culture. We used this model to study a group of breast cancer cell lines with distinct molecular subtypes (TNBC, HER2^+^, and ER^+^/PR^+^). We found that, when these breast cancer cell lines lost their attachment to the extracellular matrix, they accumulated a subtype of cancer stem cells (CSC) that expressed the surface markers of stem cells (e.g., CD44^+^CD24^−^). These stem-like CTCs also showed high expressions of hypoxia-inducible gene products, particularly the hypoxia-inducible carbonic anhydrase IX (CAIX). Inhibition of CAIX activity was found to reduce CAIX expression and stem cell phenotypes in the targeted CTCs. Further studies are needed, using CTC samples from breast cancer patients, to determine the role of CAIX in CTC survival, CSC transition, and metastasis. CAIX may be a useful surface marker for the detection of CSCs in the blood, and a potential target for treating metastatic breast cancers.

## 1. Introduction

Metastasis is the leading cause of cancer treatment failure, accounting for almost 90% of cancer deaths. In metastasis, cancer cells spread from the main tumor and move through the bloodstream or lymphatic system to form a secondary tumor in a different site. Once the cancer cells have entered the bloodstream, they travel alone or in small clusters (known as circulating tumor cells (CTCs)) until they die or settle in another spot [1]. Measuring CTCs in a patient’s blood has become a non-invasive way to check whether they are at a high risk of developing metastatic cancers [2,3] or to monitor their response to treatment [4,5]. However, accurately detecting CTCs is difficult, as the markers that are usually used to identify epithelial cells [6] are not always reliable for capturing CTCs, particularly during epithelial–mesenchymal transition (EMT). Furthermore, CTCs are rare events and often present as only a few cells per milliliter of blood. They are also highly varied, with different molecular and physical features [7,8]. Only a small subset of CTCs may survive in the bloodstream, with the majority dying from either mechanical stress, immune cell targeting, or anoikis. Understanding the molecular features of CTC subtypes can help to develop cancer diagnostic technology and new therapies to battle metastatic cancers.

In vitro CTC models can be used to overcome the low frequency of primary CTCs in the blood and allow for comprehensive characterization. These models use a non-adherent suspension culture to imitate the circulation, which allows for comparison between CTCs and cells cultured in adherent conditions (mimicking primary tumor cells). Several groups have used in vitro models to successfully propagate breast cancer CTCs and study tumor spheres after the loss of cell-matrix interactions, which revealed an increase in the stem-like CD44^+^CD24^−^ population [9,10,11]. CD44 is a marker for cancer cell migration and is involved in the extravasation of tumor cells into the bloodstream [12,13]. CD24 is an epithelial differentiation marker that is reduced during EMT and is found in low or negligible levels in cancer stem cells [14,15]. CD44^+^CD24^−^ cell populations have been found to have the greatest tumor-initiating potential in breast cancer [1,11] and have been used, either alone or in combination with other markers (CD133, EpCAM, CD49f, CD90, and CD61) [1,3], to identify and isolate breast cancer stem cells (BCSCs).

This study aimed to characterize the molecular features of circulating breast cancer stem cells (cBCSCs) using relevant CTC models. We tested a panel of five breast cancer cell (BCC) lines that represented the molecular subtypes of triple-negative breast cancer (TNBC), HER2^+^, and ER^+^/PR^+^, including MB-231, BT20, MCF7, T47D, and ZR75-1. By culturing the cell lines in suspension and adherent conditions, we were able to compare the molecular changes when the cells lost their attachment to the matrix. We made a novel observation that the expression levels of the hypoxia-inducible factor 1 (HIF1)-regulated protein carbonic anhydrase 9 (CAIX) increased significantly in suspension-induced CTCs, particularly in the CD44-positive cell population. Furthermore, the pharmacological inhibition of CAIX was effective in reducing the CD44^+^CD24^−^ population. Our data suggest a close link between hypoxia and stem-like phenotypes in circulating breast cancer cells. Our findings warrant further studies to determine the roles of hypoxia-inducible genes (e.g., CAIX) in metastatic breast cancer.

## 2. Materials and Methods

### 2.1. Cell Lines and Culture Conditions

A panel of human breast cancer cell (BCC) lines was obtained from the American Type Culture Collection (ATCC, Manassas, VA, USA) and cultured according to the supplier’s instructions. Cell lines were chosen based on their marker status (estrogen receptor-positive (ER+), progesterone receptor-positive (PR+), human epidermal growth factor receptor 2-positive (HER2+), or triple-negative breast cancer (TNBC), including TNBC basal-like (TNBC-BL) or claudin-low (TNBC-CL) cell lines (Appendix A, Appendix A) [16]. HCC1937 (CRL-2336), HCC38 (CRL-2314), ZR75-1 (CRL-1500), and AU565 (CRL-2351) were cultured in RPMI-1640 Medium (ATCC, 30-2001), supplemented with fetal bovine serum (FBS) to a final concentration of 10%. BT20 (HTB-19) cells were cultured in Eagle’s Minimum Essential Medium (ATCC, EMEM) (30-2003) supplemented with FBS to a final concentration of 10%. MB-231 (HTB-26) cells were cultured in DMEM/F-12 (1:1) medium (Mediatech, Herndon, VA, USA), containing 5% FBS, 4 mM of glutamine, 50 µM of β-Mercaptoethanol, and 1 mM of sodium pyruvate. The HS578T (HTB-126) cell line was cultured in Dulbecco’s Modified Eagle’s Medium (DMEM) (ATCC, 30-2002) supplemented with 0.01 mg/mL of human insulin (Gibco, ThermoFisher Scientific, Carlsbad, CA, USA, 12585-014) and 10% FBS. BT549 (HTB-122) cells were cultured in RPMI-1640 Medium supplemented with 0.023 U/mL of insulin and 10% FBS. T47D (HTB-133) cells were cultured in RPMI-1640 medium supplemented with 0.2 U/mL of human insulin (Life Technologies, ThermoFisher Scientific, Carlsbad, CA, USA; 12585-014) and 10% FBS. MCF7 (HTB-22) cells were cultured in EMEM supplemented with 10% FBS and insulin. BT474 (HTB-20) cells were cultured in Hybri-Care Medium (ATCC, 46-X) supplemented with 1.5 g/L of sodium bicarbonate and 10% FBS. SKBR3 (HTB-3) cells were cultured in McCoy’s 5a Medium (30-2007) supplemented with 10% FBS. UFH-001 (Sigma Aldrich, St. Louis, MO, USA; SCC210) cells were obtained from Sigma Aldrich and cultured in DMEM with 2 mM of L-glutamine and 10% FBS. All cell lines were cultured as recommended by the supplier and maintained at 37 °C with 5% CO_2_, with media changes every other day. Suspension culture was performed in Corning Ultra-Low Attachment plates (Corning, New York, NY, USA), with media changes every other day, for up to 14 days.

### 2.2. Anoikis Screening of Suspension Cultured Cell Lines

Suspension-induced anoikis was analyzed using flow cytometry. Cells were cultured in monolayer or suspension conditions for up to 14 days. Cells were collected and dissociated using Cell Stripper, a non-enzymatic dissociation buffer (Corning^TM^, Corning, New York, NY, USA), and washed with 1× phosphate-buffered saline (PBS). Cells were stained using Annexin V-FITC (BD Biosciences, Franklin Lakes, NJ, USA) and propidium iodide (PI). Cells were washed with PBS and re-suspended in 200 µL of PBS and then analyzed on an Accuri C6 flow cytometer (BD Biosciences). Gating was performed on un-stained control cells.

### 2.3. Detection of Breast Cancer Stem Cell Populations

Surface expression levels of cancer stem cell markers CD44, CD24, and CAIX were measured using multispectral imaging flow cytometry (MIFC). Cells were cultured in either monolayer or non-adherent suspension culture for up to 14 days. Cells were dissociated using Cell Stripper, homogenized into single-cell suspensions, and washed twice with cold 1× PBS. Cells were resuspended to a concentration of 5 × 10^6^ cells/mL in a blocking solution (1.0% bovine serum albumin and 5.0% normal goat serum (Invitrogen, ThermoFisher Scientific, Carlsbad, CA, USA) in 1× PBS) and kept on ice for 20 min to inhibit non-specific antibody labeling. Cells were labeled for 45 min in the dark on ice using an anti-CD44 monoclonal antibody conjugated to phycoerythrin (PE) (R&D Systems, Biotechne, Minneapolis, MN, USA; FAB6127P), an anti-CD24 monoclonal antibody conjugated to violet blue (Miltenyi Biotec Inc., Auburn, CA, USA; 130-126-026), an anti-CAIX conjugated to AF488 (R&D Systems, FAB2188G), or an anti-EpCAM monoclonal antibody conjugated to PE (R&D Systems, FAB9601P). Cells were labeled for single target samples or in indicated combinations, according to the manufacturer’s instructions, using antibody concentrations determined from titer optimization studies. IgG isotype controls for each antibody labeling and subtype were used as non-specific labeling controls. Cells were washed twice and resuspended in PBS for analysis using the MIFC. In each experiment, a minimum of 50,000 cells were acquired using a 12-channel Amnis^®^ FlowSight (Luminex, Austin, TX, USA), imaging flow cytometer, equipped with 405 nm and 488 nm lasers. Samples were acquired at 40× magnification. Single-color controls were also acquired for compensation analysis to minimize spectral spillover. IDEAS^®^ (Amnis Ideas 6.2, EMD Millipore, St. Louis, MO, USA), software was used for data collection and analysis. Single-cell populations were determined using a bivariate plot of the aspect ratio and the cell area. A gradient RMS of the bright field images was used to identify the in-focus cells. Surface expression was determined from the median fluorescence intensity (MFI) of the target protein relative to the corresponding isotype control (relative MFI, rMFI). Cancer stem cell (CD44^+^CD24^−^) or CAIX-positive (CAIX^+^) subpopulations were determined from bivariate charts, which were gated on the corresponding isotype controls. For U-104 (SLC-0111) treatment, cells were treated with indicated concentrations for 24 h. Cells were then washed with 1× PBS prior to harvest with Cell Stripper and analysis for CAIX expression. For CoCl_2_ treatment, cells were treated with 200 μM of CoCl_2_ for the indicated time points. For CoCl_2_ and U-104 combination treatment, the CoCl_2_-containing medium was removed, and cells were washed 1×. Fresh medium, containing the indicated concentration of U-104 and 200 μM of CoCl_2_, was incubated for the indicated time points—at which point, the medium was removed, the cells were washed with PBS (1×), and they were harvested for immunoblotting analysis.

### 2.4. Immunoblotting

Western blot analysis was performed as previously described [17,18]. BCCs were cultured in either monolayer or suspension conditions for up to 7 days. Cells were harvested at the indicated time points and lysed using a radioimmunoprecipitation assay (RIPA) lysis buffer with a protease inhibitor. A bicinchoninic acid (BCA) protein assay (Pierce, Rockford, IL, USA) was used to determine protein concentrations, and equal amounts of lysis were resolved using electrophoresis on 4–12% NuPAGE Bis-Tris gels and transferred to PVDF membranes. Membranes were incubated overnight with antibodies specific to human proteins (obtained from R&D Systems: CD44 (AF3669) and SNAIL (AF3639); Cell Signaling Technologies: Sox2 (14962), GLUT1(12939s), ENO2 (65162s), CX43 (3512s), PARP (9542s), and EpCAM (2626s); Santa Cruz Biotechnology: CD24 (sc-58999), CAIX (sc-365900), CAXII (sc-374314), SLC6a6/TAUT (sc-393036), and TWIST (sc-15393); Novus Biologicals: GAPDH (2D4A7) (NB300-328); BD Biosciences: HIF1α (610958), N-Cadherin (610920), and E-Cadherin (610404)). Primary antibodies were used at the manufacturer’s recommended indications (1:500 to 1:1000). Membranes were stripped using Restore Western Blot Stripping Buffer (Pierce) and re-probed with appropriate antibodies. Immunocomplexes were visualized with chemiluminescence. Densitometry analysis was performed using a LAS-4000 Luminescent Image Analyzer (Fujifilm, Lexington, MA, USA). Band density was quantified with Image J software (Version 1.51, 23 April 2018; http://imagej.nih.gov).

### 2.5. Gene Expression Analysis

Human OneArray^®^ Plus (HOA version 6.2, Phalanx Biotech Group, Inc., San Diego, CA, USA) was used for gene expression analysis, as previously described [17]. MB-231 and MCF7 cells were cultured in monolayer or suspension for 7 days prior to harvest. RNA was then extracted, and RNA quality was assessed using a Nanodrop ND-1000. Pass criteria of the absorbance ratios, of A260/A280 ≥ 1.8 and A260/A230 ≥ 1.5, were used for RNA quality control. The RNA integrity number (RIN) pass criterion of >6 was used to determine the acceptable RNA integrity. Gene expression fold changes were calculated by the Rosetta Resolver 7.2, with an error model adjusted by the Amersham Pairwise Ratio Builder. The differential expression of genes was determined through the selection criteria of log2|fold change| ≥ 1 and *p* < 0.05. The data shown are the log2 ratios (suspension compared to monolayer) of each cell type with the corresponding *p*-value. Gene expression patterns were analyzed to determine overlapping genes that were significantly up- or downregulated in both MCF7 and MB-231 cells. The predicted localization of the proteins from the identified molecular signature was analyzed using Uniprot (https://www.uniprot.org/, accessed on 22 January 2020). The list of genes was also analyzed using The Database for Annotation, Visualization, and Integrated Discovery (DAVID) (Version 6.8, david.ncifcrf.gov, accessed on 22 January 2020). Using DAVID, GO enrichment analyses for upregulated and downregulated genes were performed separately against the H. sapiens proteome. Microarray data are available at the Gene Expression Omnibus and are accessible through GEO Series accession number GSE224841.

### 2.6. Statistical Analysis

GraphPad Prism 8 (GraphPad Prism version 8.3.0 for Windows, GraphPad Software, San Diego, CA, USA) was used to perform one-way ANOVA tests and unpaired *t*-tests with a Welch’s correction on data for cell viability, flow cytometry surface expression, relative immunoblot densitometry values, and population analysis. All assays were completed with an N ≥ 3. Statistical significance is shown as either * *p* < 0.05 or ** *p* < 0.01, as indicated in the results.

## 3. Results

### 3.1. Non-Adherent Suspension Condition Enriches for a Circulating Cancer Stem Cell Population

We analyzed a panel of BCC for stem-cell and EMT-related markers, using immunoblotting, to identify those with an emergent cCSC population (Appendix A). TNBC cell lines showed a high level of CD44, while ER^+^/PR^+^ and HER2^+^ cell lines showed a high level of CD24 expression. EpCAM expression, which decreased during EMT, showed varying levels across the panel. We then cultured these cell lines in either a monolayer or suspension condition for up to 14 days to identify those cell lines with anoikis-resistant subpopulations. One of the primary mechanisms of metastatic spread is anchorage-independent cell growth and survival in the bloodstream following their release from a primary tumor [19]. Cells lines under the suspension condition underwent either a grapelike formation, with loose connections between clusters of cells, or a spheroid formation (Appendix A) [20,21]. Cell lines that maintained a viability level of over 50% during the suspension culture, with no PI or annexin V detection, throughout the 14 days were determined to be anoikis-resistant. Moving forward, the duration of 7 days of suspension culture was chosen to analyze cells prior to necrotic core formation in the spheroid-forming cell lines. We identified seven anoikis-resistant cell lines for the analyses of CSC surface markers (Appendix A).

Suspension culture is a commonly used in vitro method for propagating CSCs as tumor spheres [9,10,11]. We analyzed each cell line for the CSC marker combination of CD44 and CD24 (Figure 1) as well as the EMT marker of EpCAM (Appendix A). CD44 total protein expression increased over the culture’s time in the suspension condition, compared to the respective monolayer, in the MB-231, MCF7, T47D, and ZR75-1 cell lines, with associated reductions in CD24 in the BT20, MCF7, ZR75-1, and T47D cell lines (Figure 1A and Appendix A). Cells with low basal levels of CD24 showed no change in expression. Sox2 total protein expression, a stem cell transcription factor, also increased over the seven days in the suspension culture in MB-231, T47D, MCF7, and ZR75-1 cells. Variable expression of the differentiation factors of CD24 and Sox2 was seen in the spheroid-forming cell lines, which was attributed to the changes in the cell–cell and cell–microenvironment interactions [22]. The CD44^+^CD24^−^ population, determined using surface protein expression, increased the most in the BT20, T47D, and MCF7 cell lines under suspension compared to their respective monolayer cultures (Figure 1B–D). The EpCAM total and surface presence was reduced by day 7 in suspension culture, with associated decreases in the epithelial marker of E-Cadherin and increases in the more mesenchymal markers of N-Cadherin, Vimentin, SNAIL, and TWIST (Appendix A). E-cadherin expression is associated with reduced migration in tumor cells, which may be downregulated by SNAIL and TWIST transcription regulators [23]. Overall, suspension culture enriched the stem-like population across all the BCC lines tested when compared to their counterparts in monolayer cultures.

### 3.2. Identification of a Molecular Signature of Circulating Breast Cancer Stem Cells

To identify a transcriptomic profile of cCSCs across breast cancer subtypes, a DNA microarray (HumanOne Array) was performed on RNA samples prepared from MCF7 and MB-231 cells grown in monolayer or suspension culture for seven days [17]. Principal component analysis (PCA) plotting was performed, confirming the reproducibility of the gene expression patterns with the clustering of the replicates and the separation of the monolayer and suspension cultures per cell line (Figure 2A). Gene expression was narrowed down to only analyze those genes with significant dysregulation between day 7 (suspension culture) relative to Day 0 (monolayer culture) (log2|fold change (suspension (D7)/monolayer (D0))| ≥ 1, *p* < 0.05), which are shown in the volcano plot (Figure 2A,B) [24]. The expression profile of the impacted genes following suspension culture was then compared between MCF7 and MB-231 cells, identifying 102 common genes. These were discriminated to examine the genes that were upregulated (55 genes) or downregulated (33 genes) by suspension culture in both cell lines. The upregulated and downregulated gene profiles were analyzed separately, using DAVID to identify the most impacted signaling pathways (Appendix A). The response to hypoxia, the positive regulation of angiogenesis, canonical glycolysis, extracellular matrix reorganization, and HIF-1 signaling were the most significantly impacted gene ontology (GO) biological pathways and KEGG pathways in the upregulated gene expression panel, indicating that the suspension culture activated hypoxia-induced metabolic reprogramming in both cell lines (Appendix A). CSCs are reported to secrete angiogenic factors, aiding the pre-metastatic niche [25]. The analysis of the downregulated genes showed that the innate immune response pathway was significantly impacted, which may indicate that the suspension-induced CTCs may evade immune surveillance and potentially inhibit immune targeting.

As the suspension-cultured cells had a distinct profile relative to the monolayer controls, we further examined the common upregulated gene expression for the predicted protein localization using Uniprot (Table 1). Overall, 33 out of the 55 upregulated proteins were predicted to localize to the plasma membrane. The theoretical cell surface protein panel consisted of proteins associated with cell–cell junctions (GJA1 and RND1) and those that survive in hypoxic conditions (CA9, GLUT1, and SLC6A6). Studies have shown that CAIX is overexpressed in hypoxic tumors and may co-localize with GLUT1, a HIF1α-induced nutrient transporter that is being studied in liquid biopsies for certain types of breast cancer [9,26,27,28].

To confirm CAIX protein expression, its total and surface levels were then analyzed, using immunoblotting and flow cytometry, for the five cell lines that exhibited the highest cCSC populations on day 7 (Figure 3A,B). Immunoblotting showed that, of the identified proteins, CAIX was consistently upregulated in the five cell lines on day 7 relative to its monolayer control. As CAIX activity may be supplemented by CAXII activity in hypoxic conditions, the expression of CAXII was also studied. CAXII is not expressed in all cell lines and did not show consistent impacts of suspension culture, with reductions in expression in the MCF7 cells following 7 days of suspension culture but increases in T47D and ZR75-1. Similar results have been reported for CAXII expression in MCF7 cells under hypoxic conditions [29,30]. The role of CAXII under hypoxic conditions is still being evaluated due to its inconsistent expression and activity levels across cell lines relative to CAIX [30,31]. HIF1α total protein expression was also examined, with increases at day 7 in four of the cell lines. BT20, MB-231, MCF7, and ZR75-1 cell lines reported high CAIX^+^ populations (as a percentage of the total) with CAIX surface expression.

### 3.3. CAIX Expression Is Upregulated in the CD44^+^ Population, and the Inhibition of CAIX Reduces the cCSC Population

To determine whether there was a correlation between CAIX expression and CSC markers, CAIX expression was examined in each subpopulation of CD44^+^CD24^−^, CD44^+^CD24^+^, CD44^−^CD24^+^, and CD44^−^CD24^−^ (Figure 4A). CAIX expression was highly correlated with CD44^+^ populations, with significant increases during suspension culture in the CD44^+^CD24^+^ populations in the BT20, MCF7, and ZR75-1 cell lines and the CD44^+^CD24^−^ populations in the BT20, MB-231, T47D, and MCF7 cell lines relative to their monolayer (D0) controls (Figure 4B). As CAIX is necessary for hypoxia-induced CSCs to form tumor spheres [9,32], we studied the impact of culturing the cCSC cell lines with a small molecular inhibitor, Ureido-substituted benzenesulfonamide (U-104/SLC-0111). U-104 inhibits the extracellular catalytic domain of CAIX and is being developed for the treatment of advanced solid tumors (NCT02215850) [33] and metastatic pancreatic ductal cancer in patients positive for CAIX (NCT03450018), either as a single agent or in combination with chemotherapy. U-104 not only inhibits the CAIX activity but is able to reduce total CAIX expression [34]. To determine a subtoxic dosage of U-104, UFH001 cells, which constitutively express CAIX, were treated with U-104 for up to 72 h (Appendix A). UFH-001 cells underwent cell death by 72 h of treatment, as seen by the significant cleavage of PARP. CAIX and CAXII decreased expression in a time- and dose-dependent manner up to 48 h. Cobalt chloride (CoCl_2_) treatment was used as a chemical-induced model of hypoxia in MB-231 cells, due to the prevention of HIF1α degradation, to determine the impact of CAIX on the viability and expression of CD44 and CAIX in the monolayer (Appendix A). CoCl_2_-induced CAIX expression was effectively inhibited by U-104 after 24 h of treatment without the cleavage of PARP. CD44 expression increased slightly following CoCl_2_ treatment, which was not impacted by U-104 in the monolayer condition.

Using our CTC model, cells were initially cultured in a suspension or monolayer for 3 or 7 days, and then cultured with fresh media containing U-104 for 24 h at a low (10 µM), medium (50 µM), or high (100 µM) concentration. The low and medium concentrations did not significantly impact viability, while the high concentration reduced viability in the monolayer-cultured cells (Appendix A). The inhibition of CAIX has been reported to reduce the proliferation rates of BCCs in monolayer cultures [29]. Cells were analyzed for the surface expression of CAIX, CD44, and CD24 using flow cytometry, and the positive populations were quantified. The inhibition of CAIX activity significantly reduced CAIX^+^ and CD44^+^CD24^−^ populations in MB-231 and MCF7 cells at each time point (Figure 5). CAIX activity appears necessary for the cCSC population to emerge under a suspension culture, though inhibition does not impact anoikis resistance in the suspension-conditioned cells. Together, these results suggest that CTCs may acquire stem-like phenotypes through hypoxia-induced CAIX expression, leading to anoikis resistance and metastasis potential.

## 4. Discussion

Defining the molecular markers on CTC populations that span the molecular subtypes of breast cancer has the potential to improve breast cancer diagnosis and to identify novel therapeutic targets for treating metastatic cancers. However, this effort is often hindered by CTCs’ low frequency in blood, vast heterogeneity, poor survival, and analytical variability in isolating the cells. To address this, improved methods for CTC culture and expansion are necessary to facilitate comprehensive CTC studies. Several CTC models have been developed, including 2D cultures and CTC-derived explants (CDX models) using primary CTCs isolated from the blood of patients [35]. Here, we established an in vitro model of CTCs using non-adherent suspension cultures of BCC lines representing the molecular subtypes of TNBC, HER2^+^, and ER^+^/PR^+^. In our design, individual BCC lines were cultured in suspension and under a monolayer, in parallel, thereby allowing for the pairwise analysis of CTCs and their counterparts in primary tumors. Overall, this CTC model was able to recapitulate several major molecular alterations in CTCs, as observed previously using primary CTCs from breast cancer patients [36,37,38]. In particular, the non-adherent suspension culture enriched CSC populations upon losing cell-matrix adhesion. Enriched CSCs were further shown to express higher levels of hypoxia-inducible genes—in particular, the hypoxia-inducible carbonic anhydrase IX (CAIX)—when compared to the monolayer counterparts. Moreover, the pharmacological inhibition of CAIX effectively reversed the stem-like phenotypes. The data draw a link between hypoxia-inducible genes (e.g., HIF1, CAIX) and stemness transition in CTCs (Figure 6).

CAIX can be used to help isolate circulating cancer stem cells (CSCs) from the blood samples of patients with metastatic breast cancer, making it a potential target for treatment. Current studies are attempting to use CAIX to isolate renal cell carcinoma CTCs [39]. Our CTC model selects for anoikis-resistant cancer cells using both tumor sphere-forming and non-tumor sphere-forming cell lines, as primary CTCs exist in the body as both single cells and clusters [40,41,42]. Both tumor sphere-forming cell lines, such as BT20, T47D, and MCF7, and non-tumor sphere-forming cell lines, such as MB-231 and ZR75-1, were found to be anoikis-resistant. We have previously shown that CTCs grown in vitro can survive anoikis and increase cell survival pathways, including autophagy, which leads to the breakdown of pro-apoptotic cell surface receptors [17]. The anoikis-resistant cell lines had suspension-induced increases in the total protein expression of CSC marker CD44 and the EMT-associated proteins of N-Cadherin, vimentin, SNAIL, and TWIST, with the associated reductions in the EpCAM total and surface expression. Sox2, a stem-cell transcription factor, was upregulated over the course of the 7 days in suspension culture relative to monolayer controls, which is necessary for the tumor sphere formation of BCC lines [43]. cCSCs are known to have stem-like properties that are either induced or selected by the harsh tumor microenvironment in the bloodstream. The CD44^+^CD24^−^ population was consistently high in the MB-231 cell lines, with significant increases in the BT20, T47D, and MCF7 lines.

CAIX is a cell surface metalloenzyme with an extracellular active domain that helps maintain the balance of intracellular and extracellular pH by reversibly hydrating extracellular carbon dioxide [27,29]. This can lead to acidosis of the tumor microenvironment, the loss of cellular adhesion, and increased tumor cell migration [44]. Cancer cells often undergo metabolic reprogramming due to oxygen and nutrient deprivation within solid tumors, resulting in an upregulation of CAIX. Studies have shown that CAIX is a key hypoxia-inducible protein marker, and higher levels of CAIX are associated with poorer survival in patients with TNBC and ovarian, bladder, and lung cancers [45,46]. Additionally, the depletion of CAIX^+^ cells has been shown to reduce the number of cancer stem cells (CSCs) in primary solid tumors and cancer cells in the bloodstream (CTCs) [9]. Interestingly, CAIX was upregulated in five BCC cell lines following suspension culture, with the highest expression seen in the TNBC lines. In contrast, CAXII, which has similar activity to CAIX, decreased slightly in the MCF7 cell lines after seven days of suspension relative to the monolayer. The analyses of CTCs from metastatic breast cancer patients have consistently shown high levels of HIF1α expression, though levels of CTC-CAIX have yet to be studied [47]. CAIX is also being evaluated as a potential therapeutic target [48]. Clinical studies of anti-CAIX monoclonal antibodies or small molecule inhibitors have shown promising results, though the associated toxicity still needs to be addressed [33,34,49,50].

Our CTC model showed decreases in both CAIX^+^ and CD44^+^CD24^−^ populations when treated with sub-toxic doses of U-104. After examining gene expression, we noted changes in the innate immune response, which is consistent with previous research that found reduced immune cell activity in tumor areas with high CAIX expression [51,52]. These findings support the usefulness of our suspension culture model for studying CTCs and suggest that CAIX could be a viable target for treating metastatic breast cancers.

## 5. Conclusions

A suspension culture that imitates blood circulation and does not involve cell-matrix adhesion is a useful tool for examining the molecular changes of CTCs. This lack of cell-matrix adhesion stimulates the expression of hypoxia-induced genes, such as CAIX, and increases the number of cCSCs across a range of breast cancer cell lines. CAIX can be used as a surface marker to help detect cCSCs in the blood and may potentially be a new therapeutic target for treating metastatic breast cancer.

## Figures and Tables

**Figure 1 biomedicines-11-00934-f001:**
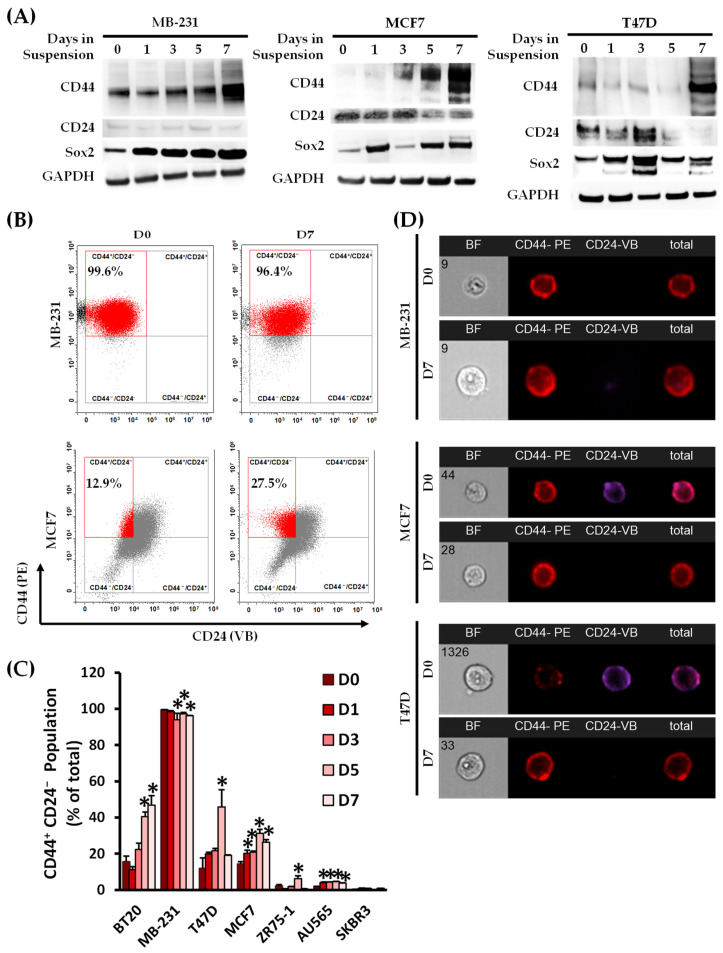
Identification of a panel of breast cancer cell lines with emergent anoikis-resistant cancer stem cell populations. (**A**) A panel of seven breast cancer anoikis-resistant cell lines was evaluated for cancer stem cell markers CD44, CD24, and Sox2 following culturing in either a monolayer (day 0) or non-adherent suspension condition for up to seven days using immunoblotting. Total protein expression for CD44 and Sox2 was increased over the seven days of culturing relative to the monolayer condition. CD24 was decreased in the MCF7 and T47D cell lines, and low expression was maintained in the MB-231 cell line. Immunoblot images are representative of 3 biological replicates. (**B**) The surface expression of CD44 and CD24 was analyzed using multispectral imaging flow cytometry (MIFC), and the CD44^+^CD24^−^ population was quantified from the bivariate graph (**C**). The data shown are the percentages of CD44^+^CD24^−^ cells out of the total population. Bivariate charts were gated on the isotype controls for each marker. (n = 3, * *p* < 0.05 relative to the monolayer culture (D0)). (**D**) MIFC images were taken of individual cells at each time point, and representative images for MB-231, MCF7, and T47D cells at days 0 (D0) and 7 (D7) of suspension culturing are shown. Assigned cell number of population is shown on the brightfield images.

**Figure 2 biomedicines-11-00934-f002:**
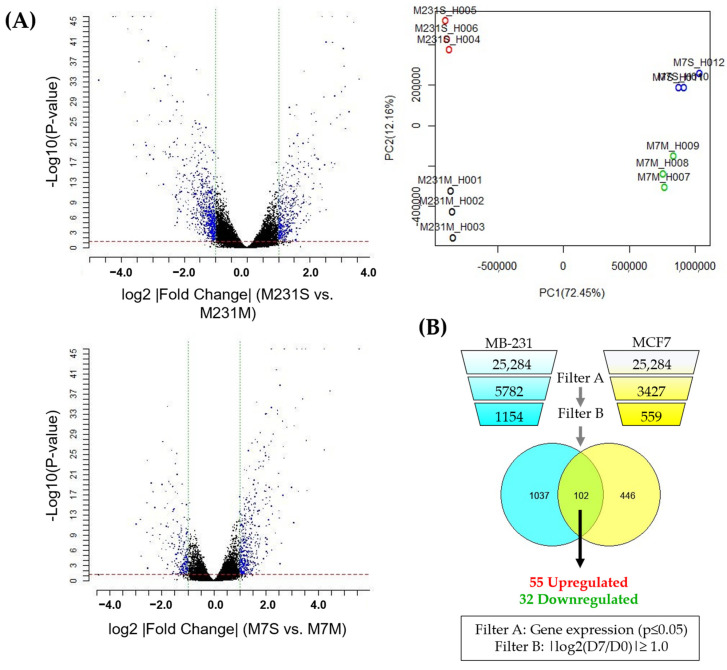
Molecular signature of suspension-cultured breast cancer cell lines using Gene microarray. (**A**) MB-231 and MCF7 cells were collected from the monolayer following 7 days of suspension culture and analyzed using the Human One Array (Phalanx Biotech). The volcano plots for MB-231 suspension and monolayer (M231S/M231M) and MCF7 suspension and suspension (M7S/M7M) and principal component analysis (PCA) plot for these analyses are shown (PC1 84.37%, PC2 9.17%, PC3 3.90%). (**B**) Differentially expressed genes at 7 days of the suspension culture (D7) relative to the monolayer (D0) were established using the standard selection criteria of log2 |fold change| ≧ 1 and a *p*-value < 0.05 (shown in the volcano plot as blue dots). (M231M, MB-231 cells subcultured in monolayer; M231S, MB-231 cells subcultured in suspension for 7 days; M7M, MCF7 cells subcultured in monolayer; M7S, MCF7 cells subcultured in suspension for 7 days). Normalized gene expressions of suspension-cultured cells relative to the monolayer were narrowed to look at significantly regulated genes (gene expression change *p* < 0.05; filter A), which were either up or downregulated by a fold of 50% (|log2(D7/D0)| ≥ 1.0, filter B). These gene lists were analyzed to determine a list of common genes between the two cell lines, which were then discriminated based on whether they were upregulated or downregulated in both cell lines. This analysis resulted in about 102 shared genes, of which, 55 were upregulated and 32 were downregulated.

**Figure 3 biomedicines-11-00934-f003:**
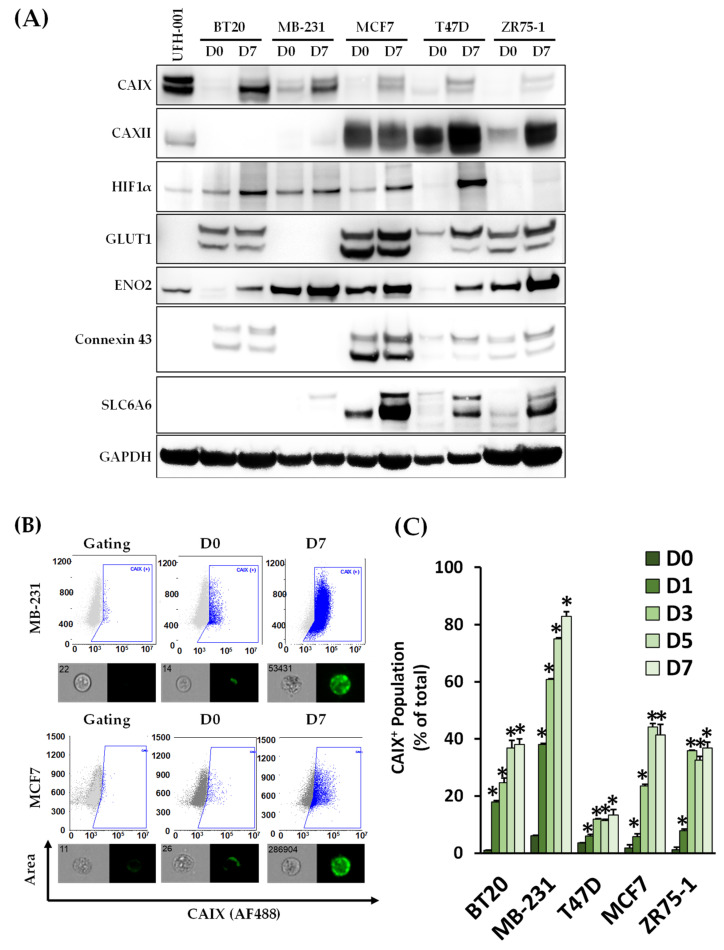
Suspension-cultured breast cancer cell lines expressed high levels of carbonic anhydrase IX. (**A**) Breast cancer cell lines were collected from the monolayer (D0) following 7 days of suspension culture (D7) and analyzed using immunoblotting for markers identified in the top dysregulated cell surface proteins as well as HIF1α. Carbonic Anhydrase IX (CAIX) and ENO2 were significantly upregulated following suspension culture in the cell lines studied. (**B**) The surface expression of CAIX was detected using MIFC with gating performed on the isotype and unlabeled controls for each cell line. (**C**) CAIX surface expression increased in each cell line over the 7 days of culture, with the most prominent expression seen in MB-231, MCF7, and ZR75-1 cells. CAIX^+^ populations were quantified at each indicated time point. (n = 3, * *p* < 0.05, relative to monolayer culture).

**Figure 4 biomedicines-11-00934-f004:**
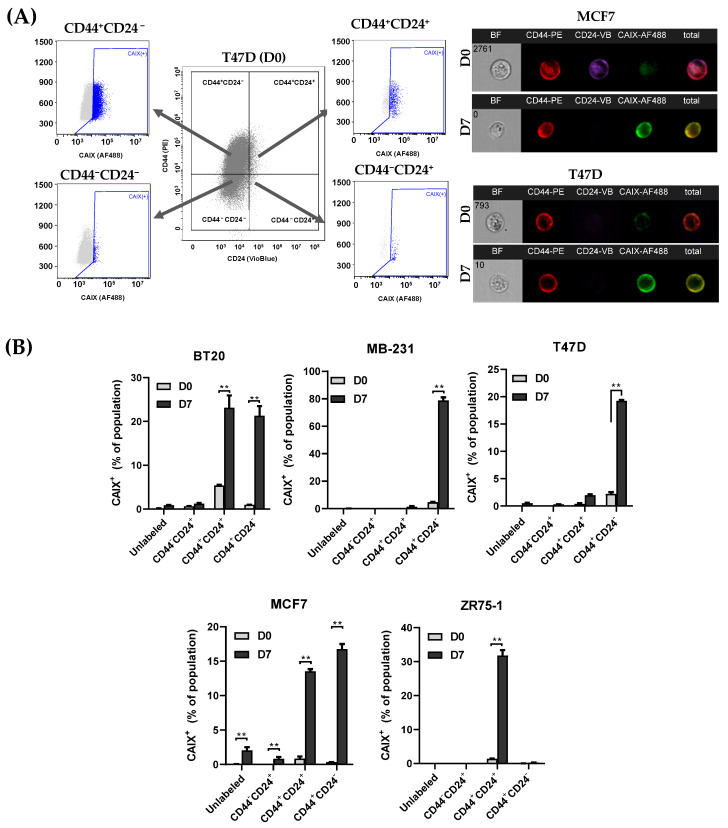
Surface expression of Carbonic Anhydrase IX (CAIX) is predominantly expressed in the CD44-positive population. (**A**) BCCs were cultured in a monolayer (D0) or suspension condition (D7) and then analyzed for CD44, CD24, and CAIX using MIFC. (**B**) The population of CAIX^+^ cells was quantified for each subpopulation of CSC markers (unlabeled, CD44^−^CD24^+^, CD44^+^CD24^+^, and CD44^+^CD24^−^). Gating was performed on isotype controls for each marker. The largest increases in CAIX^+^ populations were seen in the CD44^+^ cells following suspension culture. n = 3, ** *p* < 0.01.

**Figure 5 biomedicines-11-00934-f005:**
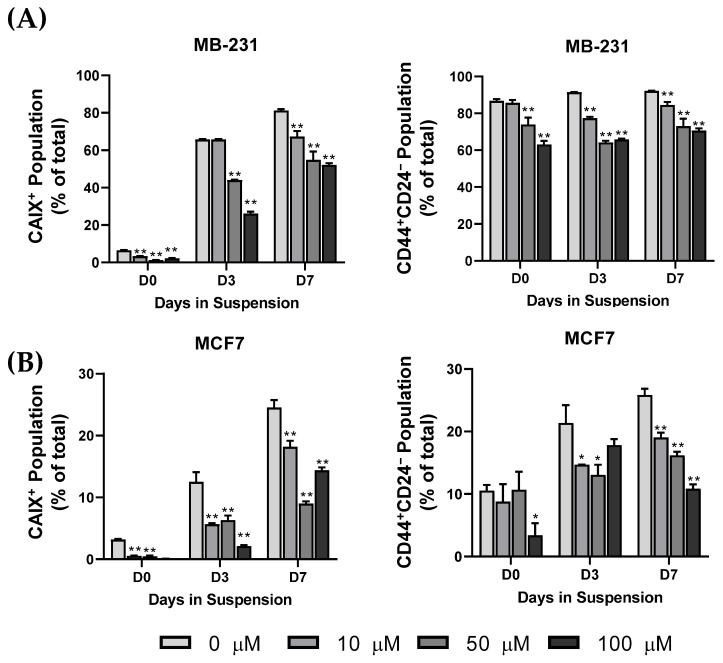
The cBCSC and CAIX populations are inhibited following treatment with U-104 under suspension. (**A**) MB-231 and (**B**) MCF7 cells were cultured in a monolayer (D0) or suspension condition for 3 or 7 days and then treated with 0, 10, 50, or 100 µM of U-104 for 24 h. Treated cells were then analyzed for CD44, CD24, and CAIX surface expression using MIFC. The CAIX^+^ and the CD44^+^CD24^−^ decreased in a dose-dependent manner with U-104 treatment. (n = 3, * *p* < 0.05, ** *p* < 0.01).

**Figure 6 biomedicines-11-00934-f006:**
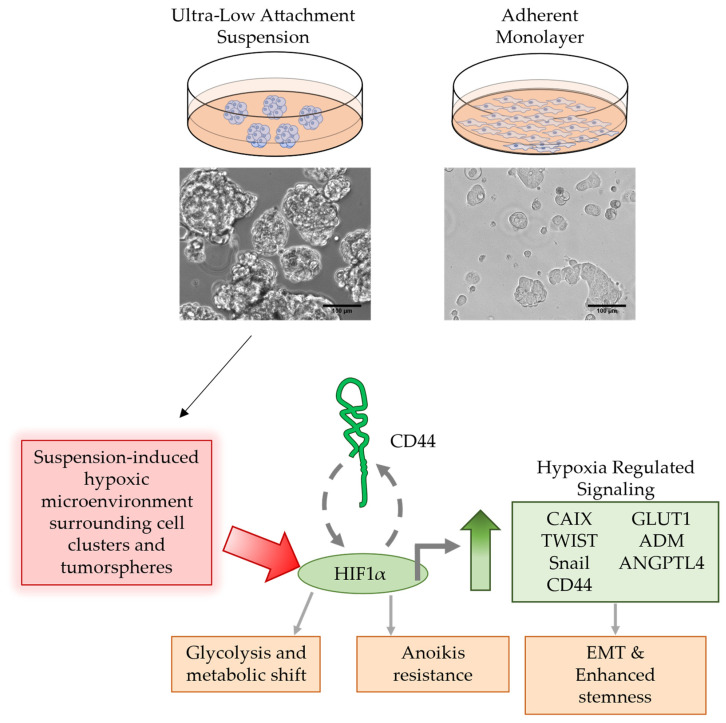
Suspension culture induces hypoxia-regulated signaling in tumor cell clusters and spheres. BCCs cultured in a non-adherent suspension condition may result in a hypoxic microenvironment surrounding the “circulating” tumor spheres and clusters. The activation of hypoxia-regulated signaling through HIF1α resulted in increases in CAIX, TWIST, Snail, and GLUT1 expression, which are known EMT factors that result in enhanced stemness of the BCSC. CD44 and HIF1 act in a positive feedback loop through PI3K/AKT signaling, resulting in sustained CD44^+^ cells through the 7 days of culture. CAIX is a potential marker that may be used to monitor or act as a therapeutic target for hypoxia-enriched cCSCs.

**Table 1 biomedicines-11-00934-t001:** Predicted protein localization of the top 10 genes of the 55 upregulated common genes, using Uniprot to identify plasma membrane-localized proteins.

Gene Symbol	Description	log2 (Ratio) (D7/D0)		Uniprot Subcellular Localization
		MB-231	MCF7	
CA9	Carbonic anhydrase IX	1.380	5.581	Plasma Membrane Nucleus
DTNA	Dystrobrevin, alpha	1.453	1.696	Plasma Membrane Cytoplasm and Cytosol
ENO2	Enolase 2 (gamma, neuronal)	1.481	2.485	Plasma Membrane Cytoplasm and Cytosol
GJA1	Gap junction protein, alpha 1, 43 kDa; Connexin-43	1.470	1.044	Plasma Membrane Endoplasmic reticulum
IL1RAP	Interleukin 1 receptor accessory protein	1.176	1.060	Plasma Membrane Extracellular region or secreted
RGS2	Regulator of G-protein signaling 2, 24 kDa	1.268	3.077	Plasma Membrane Nucleus Cytoplasm and Cytosol Mitochondrion
RND1	Rho family GTPase 1	1.073	1.080	Plasma Membrane Cytoskeleton
SLC2A1/GLUT1	Solute carrier family 2 (facilitated glucose transporter), member 1	1.163	2.025	Plasma Membrane
SLC6A6	Solute carrier family 6 (neurotransmitter transporter, taurine), member 6	1.493	1.696	Plasma Membrane
UBE2C	Ubiquitin-conjugating enzyme E2C	1.486	1.274	Plasma Membrane Cytosol Nucleus

## Data Availability

The data presented in this study are available upon request from the corresponding author.

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
