# Peer review of "Exploring the Role of Hypoxia-Inducible Carbonic Anhydrase IX (CAIX) in Circulating Tumor Cells (CTCs) of Breast Cancer"

_biomedicines, 2023, doi:10.3390/biomedicines11030934_

Round 1

Reviewer 1 Report

In this paper, the authors explored circulating tumor cells in breast cancer. A new non-adherent suspension culture method was used to model circulating tumor cell lines while comparing cell lines in monolayer culture conditions as a control. The circulating tumor cells after suspension culture showed EMT characteristics and enriched stem cell-like characteristics. An in-depth comparison of suspended and adherent breast cancer cell lines showed that circulating tumor cells expressed the hypoxia-Inducible gene, especially CAIX. The absence of CAIX+ cells has been shown to reduce the number of cancer stem cells (CSCs) in primary solid tumors and cancer cells (CTCs) in the blood, providing a potential therapeutic target.

In this paper, the logic is clear, moving from the establishment and validation of circulating tumor cell culture models to the exploration of mechanisms to discover CAIX and provide potential therapeutic targets. The experimental design is clear and the results are reliable, suggesting that CAIX can be used as an important target for the early diagnosis and treatment of breast cancer peripheral blood.

Author Response

We appreciate the suggestion of improving the clarity of the methods and have revised to improve readability.

Reviewer 2 Report

Very good and interesting papier

Author Response

We thank the reviewer for their assessment and have revised our results section. 

Reviewer 3 Report

The manuscript describes an in vitro model of metastatic cancer using cell lines from different subtypes of breast cancer. The authors describe an increase in CD44+/C24- cells over time in suspension culture in an effort to mimic the metastatic environment for circulating tumour cells. They describe an increase in CAIX along with a related increase in HIF1alpha. The hypothesis as presented is that this change represents a reprogramming of circulating tumour cells related to hypoxia regulated signalling which leads to a metabolic shift, anoikis resistance and enhanced EMT/”stemness”  of the circulating tumour cells.

The manuscript is very well written and there are no issues with the language. Syntax and grammar is excellent. I only managed to spot one typographical error line 409 Twist should read as TWIST for conformity.

Overall this is a well written, thoughtfully designed study which has important novel findings around hypoxia and metastasis which significantly contribute to the literature. The methodology is useful for screening potential therapies.

 However, the authors need address and to provide more interpretation of the results that don’t quite fit the hypothesis as listed below. The biggest concern for me is around the number of repeats. Triplicate repeats in a quantitative setting need to be either justified statistically for power or further repeats would be required in my opinion.   

Specific questions are listed below.

1) The major comment is that the majority of results of the study are presented as triplicate repeats. Can you justify this relatively low number of repeats in terms of statistical power? The statistical power either needs to be justified in the text or further repeats are required.

2) It would be useful for a reader who is not as familiar with the cell lines used as the authors to have a table summarising the characteristics of each cell line used, i.e. their TNBC/ER/PR/HER2 status and other relevant data such as sphere producing or not. This would be very helpful for a reader to more easily pick out the cell lines in the text. The inclusion of the very useful supplementary figure S1 in the main manuscript would help in this regard in my opinion.

Certainly the original clinical correlate for the breast cancer subtypes should be mentioned in relation to their histopathology, e.g. invasive lobular carcinoma or NST (ductal) or metaplastic carcinoma as this starting point is very important in making a clinical connection with the in vitro data.

3) In Figure 1 Panel A, it states Total protein expression for CD44 and Sox2 were increased and CD24 decreased over the seven days compared to monolayer. Is the Days in Suspension 0 lane on the Western blot taken to be the monolayer culture? In that panel A MB-231 does not appear to lose CD24, it remains low throughout. For T47Dthere is an increase at day 3 which is suddenly lost at day 5. Can the authors explain why it goes up first and is then lost in some cell lines?Similarly Sox2 for MCF7 goes down on day 7. The authors in the methods mention culture out to day 14. It needs to be explained why Day 7 was chosen and what the results at day 14 were.

Also in the related supplemental figure 1A and 2A it is not clear which relates to the monolayer and which relates to non-adherent suspension conditions. Please label.

4) The identification of a molecular signature of circulating breast cancer stem cells is well designed and executed. However, it would be helpful to point out to the reader why those two cell lines in particular were chosen.

5) Figure 3A shows in general that CAIX is increased  on western blots. These are convincing. However, HiF1alpha for ZR75-1 is not convincing for a related increase. The authors need to address this in the text as to the variability.

6)The authors correctly mention that CAXII has a similar function to CAIX and show that this is variable in the cell lines examined. They need to address why that might be given the similarity in function of both.

7) Figure 5 shows the potential use of U-104 to inhibit CAIX and that this could be useful for examination in the metastatic setting. The authors describe a dose-dependent decrease. There is a convincing decrease but it is not dose dependent in MCF7 with an increase at D7 for 100 uM of U-104. And an increase at D3 for CD44+/CD24- cells in MCF7 cell lines. As mentioned earlier the fact that these results are in triplicate lessens their power.

8) The authors mention that this trend or selection toward stemness in the metastatic setting may help with angiogenesis, and a metabolic shift. They show this very well in  the supplementary data supplied. However, they mention immune evasion as well. Given the use of immunotherapy in TNBC is based on PDL1 expression can the authors comment on this. In addition, a recent paper by Juhasz et al in Applied Immunohistochemistry and Molecular Morphology described CAIX in the tumour microenvironment as suppressing T cell infiltration. This may be useful to cite in this context.

9) In the literature there is a study which has looked at the potential for CAIX and HIF1 alpha as markers in a liquid biopsy. The authors should consider citing as they mention liquid biopsy in this manuscsipt.

Peiró et al. Sci Rep. 2021 Apr 22;11(1):8724. Diagnostic potential of hypoxia-induced genes in liquid biopsies of breast cancer patients.
